# *Exo*⇔*Endo* Isomerism, MEP/DFT, XRD/HSA-Interactions of 2,5-Dimethoxybenzaldehyde: Thermal, 1BNA-Docking, Optical, and TD-DFT Studies

**DOI:** 10.3390/molecules25245970

**Published:** 2020-12-16

**Authors:** Nabil Al-Zaqri, Mohammed Suleiman, Anas Al-Ali, Khaled Alkanad, Karthik Kumara, Neartur K. Lokanath, Abdelkader Zarrouk, Ali Alsalme, Fahad A. Alharthi, Afnan Al-Taleb, Amjad Alsyahi, Ismail Warad

**Affiliations:** 1Department of Chemistry, College of Science, King Saud University, P.O. Box 2455, Riyadh 11451, Saudi Arabia; anas.alali@najah.edu (A.A.-A.); aalsalme@ksu.edu.sa (A.A.); fharthi@ksu.edu.sa (F.A.A.); afnan1taleb@gmail.com (A.A.-T.); 438203007@student.ksu.edu.sa (A.A.); 2Department of Chemistry, Science College, An-Najah National University, Nablus P.O. Box 7, Palestine; suleimanshtaya@najah.edu; 3Department of Studies in Physics, University of Mysore, Manasagangotri, Mysore 570006, India; warad@najah.edu (K.A.); lokanath@physics.uni-mysore.ac.in (N.K.L.); 4Department of Physics, School of Sciences, Block-I, JAIN (Deemed-to-be University), Bengaluru 560011, India; kk.phy21@gmail.com; 5Laboratory of Materials, Nanotechnology and Environment, Mohammed V University, Faculty of Sciences, 4Av. Ibn Battuta, Rabat P.O. Box 1014, Morocco; azarrouk@gmail.com

**Keywords:** exo⇔endo isomerism, docking, density functional theory, thermal stability, X-ray diffraction, 2,5-dimethoxybenzaldehyde

## Abstract

The *exo*⇔*endo* isomerization of 2,5-dimethoxybenzaldehyde was theoretically studied by density functional theory (DFT) to examine its favored conformers via sp^2^–sp^2^ single rotation. Both isomers were docked against 1BNA DNA to elucidate their binding ability, and the DFT-computed structural parameters results were matched with the X-ray diffraction (XRD) crystallographic parameters. XRD analysis showed that the *exo*-isomer was structurally favored and was also considered as the kinetically preferred isomer, while several hydrogen-bonding interactions detected in the crystal lattice by XRD were in good agreement with the Hirshfeld surface analysis calculations. The molecular electrostatic potential, Mulliken and natural population analysis charges, frontier molecular orbitals (HOMO/LUMO), and global reactivity descriptors quantum parameters were also determined at the B3LYP/6-311G(d,p) level of theory. The computed electronic calculations, i.e., TD-SCF/DFT, B3LYP-IR, NMR-DB, and GIAO-NMR, were compared to the experimental UV–Vis., optical energy gap, FTIR, and ^1^H-NMR, respectively. The thermal behavior of 2,5-dimethoxybenzaldehyde was also evaluated in an open atmosphere by a thermogravimetric–derivative thermogravimetric analysis, indicating its stability up to 95 °C.

## 1. Introduction

Aldehydes and ketones are key building blocks for a wide range of synthetic and natural derivatives and are used in several applications such as the Schiff base reaction [1,2,3,4]. In particular, aldehydes are commonly used for the development of effective drugs due to their several biological activities resulting from the polar HC=O group [4,5,6]. Nevertheless, improvements in density functional theory (DFT) methods have allowed the reliable theoretical application of larger molecules with even more 100 atoms in the development of new pharmaceutical agents. Therefore, DFT is currently the most powerful tool for quantum chemistry computations [7,8,9,10] and, along with X-ray single crystal analysis, has become particularly valuable for structural optimizations [9,10,11,12]. Moreover, DFT has significantly contributed in the evaluation and comparison of several experimental spectral analyses [11,12,13]. Molecular docking is also a suitable method for understanding the binding mode of drugs with DNA via, e.g., noncovalent interactions [14,15,16,17,18], and is usually applied for the design of novel drug structures. Besides, both experimental and theoretical docking studies help to explore organic and inorganic complexes as potential drug candidates [18].

A literature survey revealed that the *exo*⇔*endo* isomerization of 2,5-dimethoxybenzaldehyde at the DFT/B3LYP level of theory, as well as quantum computations that have not yet been performed. Therefore, in this study, the structure of the isomerization transition state was estimated by the QST2 method. The X-ray diffraction (XRD) structure of the *exo*-isomer was identified as the kinetically favored isomer, indicating that the structure parameters determined by XRD and DFT studies were in good agreement. To establish the intermolecular forces in the crystal lattice, the results of the Hirshfeld surface analysis (HSA) and molecular electrostatic potential (MEP) computations were compared with the experimental XRD packing results. In addition, the MEP, Mulliken and natural population analysis (NPA) charges, frontier molecular orbital (HOMO/LUMO), and global reactivity descriptor (GRD) quantum parameters were determined, while the computed electronic calculations (TD-SCF/DFT/B3LYP, GIAO-NMR, and DFT-IR were matched to UV–Vis, the optical energy gap (*E*_g_), FTIR, and ^1^H NMR experimental spectra, respectively, under identical conditions. Moreover, the *exo*- and *endo*-isomers of 2,5-dimethoxybenzaldehyde could be sufficiently reflected, docked against one DNA helix DNA through the development of two strong hydrogen bonds.

## 2. Experimental Section

### 2.1. Computational Methodology

In order to determine the optimization, Mulliken, NPA, HOMO/LUMO, GRD, TD-SCF/DFT, B3LYP-IR, NMR-DB, and GIAO-NMR quantum-chemical parameter calculations of the desired molecule in a gaseous phase have been performed using Becke’s three parameter exchange function (B3) with the Lee-Yang-Parr correlation function (LYP) with basis sets 6-311G(d,p) [19,20,21]. The DFT/B3LYP/6-311G(d,p) level of theory is found to be very suitable for pure organic compounds like the desired molecule in this study [21]. Moreover, the QTS2 computation method was applied to detect the transition state (TS) of the *exo*⇔*endo* isomerization reaction [21].

### 2.2. XRD and HSA

CrystalExplorer 3.1 was used for the HSA analysis [22] using a colorless 0.29 × 0.26 × 0.23-mm single crystal of *exo*-2,5-dimethoxybenzaldehyde. The structure was solved using the SHELXL and SHELXS programs [23]. Crystal refined parameters are illustrated in Table 1.

### 2.3. 2,5-Dimethoxybenzaldehyde Crystallization

In order to obtain suitable crystals for XRD measurements, 100 mg of commercially available 2,5-dimethoxybenzaldehyde (C_9_H_10_O_3_, Aldrich, St. Louis, MO, USA, 99.0% pure) (100 mg) was dissolved in 10-mL MeOH at room temperature. After 2 days of evaporation at this temperature, colorless crystals were slowly formed.

### 2.4. BNA Docking

Docking studies were performed using the Autodock4.2 running on an Intel(R) Core(TM) i5 CPU (3 GHz) processor with a Windows 2007 operating system, Palo Alto, California, USA. The isomer structures were prepared using ChemDraw. The docking was performed using the Gasteiger charges, the water molecules were erased, and the nonpolar hydrogen atoms were merged using AutoDock4 [24]. The X-ray crystal of PDB ID: 1BNA DNA was freely obtained from the Protein Data Bank [25].

## 3. Results and Discussion

### 3.1. XRD and DFT Structure Analysis

XRD analysis indicated that *exo*-2,5-dimethoxybenzaldehyde (**A**) was the kinetically favored isomer with a dihedral angle (τ_O1–C2–C3–C4_) of 179.95°. In contrast, the thermodynamically favored *endo*-isomer (**B**) with τ_O1__–__C2__–__C3__–__C4_ = 0° was not detected by XRD (Scheme 1). 

2,5-Dimethoxybenzaldehyde crystallized in the kinetically favored *exo*-isomer form (Figure 1a) was monoclinic, with a p21/n space group, and four molecules were crystallized in a packing unit cell (Figure 1b). In the gaseous phase, the B3LYP/6-311G(d,p)-optimized *exo*-isomer structure was consistent with the XRD experimental result of the solid state, as shown in Figure 1c and Table 2.

### 3.2. B3LYP/6-311G(d,p) Structures

The bond distances and angles determined by DFT and XRD were almost identical, as shown in Table 2 and Figure 2. Specifically, the correlation (R^2^) between the calculated/experimental bond lengths was found to be 0.9845 (Figure 2a,b) and that between the calculated and experimental angles was 0.9357 (Figure 2c,d). Slight differences were only observed, because the DFT was performed in the gaseous phase, while XRD in the solid state.

### 3.3. Exo⇔Endo Computational Isomerism 

The *exo*-kinetic isomer did not favor the coordination of a metal ion, as the carbonyl oxygen and 2-OCH_3_ oxygen atoms were in the opposite direction. Although its structure was sterically favored, it did not serve as a good OՈO bidentate ligand. In contrast, the *endo*-isomer was expected to be a good OՈO bidentate chelate ligand, as the two oxygen atoms were in the same direction. Although the geometry of the *exo*-isomer was not appropriate for ligation, the formation of a stable S6-fused metal–heterocyclic ring after isomerization was confirmed by XRD crystal analysis (Figure 3).

Therefore, here, we investigated the *exo⇔endo* isomerization based on theoretical measurements to identify the energy required to switch between the two isomers. As shown in Scheme 1, the stereochemical difference between the two isomers is controlled by a 90° single rotation around the cited C_sp2_-C_sp2_ single bond, which leads to a significant change in the τ_O1_C2_C3_C4_ dihedral angle from 180° (*exo*) to 0° (endo). Based on this change, and ignoring all the intermolecular forces in both isomers, high-level DFT/B3LYP/6-311G(d) optimization calculations were performed in the gaseous state for both isomers. Moreover, the QTS2 computation method was applied to detect the transition state (TS) of the *exo*⇔*endo* isomerization reaction (Figure 4).

Based on the energy profile of the *exo*⇔*endo* isomerization of 2,5-dimethoxybenzaldehyde (Figure 4), the *exo*-isomer energy was −574.76611449 a.u., E_exo_ = 0.0 kJ, whereas that of the *endo*-isomer was found at −574.76130900 a.u., E_endo_ = 12.62 kJ. Moreover, the TS energy was higher than that of both isomers (−574.75176736 a.u., E_TS_ = 37.66 kJ), and its structure was between the structure of the *endo*- and *exo*-isomers with a dihedral angle of 84.3°. Therefore, we demonstrated that, energetically, the stable *exo*-isomer could give the unfavorable *endo*-isomer, since the energy required for isomerization was not too high and could be easily provided by the surrounding environment or solvents.

### 3.4. Crystal Interactions and HSA Investigation

Three main H∙∙∙O hydrogen bond interactions were detected in the crystal lattice of *exo*-2,5-dimethoxybenzaldehyde molecules, while each molecule was bound to its surrounding molecules through six hydrogen bonding interactions (Figure 5a), and no other types of interactions were identified. The two shortest interactions were assigned to the C=O∙∙∙H_Me_ hydrogen-bonding interactions with a distance of 2.669 Å, forming a semi-dimer S14 supramolecular system (Figure 5b), while two C=O∙∙∙H_Me_ hydrogen bonds with 2.703 Å (Figure 5c) and two MeO∙∙∙H_ph_ hydrogen bonds with 2.702 Å (Figure 5d) could also be detected.

During HSA, four red spots were detected on the *d_norm_* surface of the computed molecule [26,27,28,29,30], which were all attributed to the formation of H∙∙∙O hydrogen bonds (Figure 6a). It should be noted that the type and the number of the hydrogen bonds identified by HSA were consistent with those detected by an XRD packing analysis (Figure 6b). In addition, the HSA 2D fingerprint plots over the computed surface molecule indicated the presence of 65.9% total hydrogen interactions, which corresponded to three types of hydrogen-bonding interactions, namely H∙∙∙H (48.3%) > H∙∙∙O (13.9%) > H∙∙∙C (3.7%) (Figure 6c).

### 3.5. MEP Analysis and Atomic Charge Populations

The MEP analysis suggested the presence of both electrophilic and nucleophilic sites on the molecule surface (Figure 7a). The carbonyl oxygen atom was indicated as the strongest nucleophile site (red), while the other oxygen atoms were less nucleophilic (yellow). Moreover, the phenyl and methyl hydrogen atoms had strong electrophilic positions (blue). These findings strongly supported the formation of H∙∙∙O hydrogen bonds [31], as already confirmed by the XRD experimental results and HSA computations.

The determination of the Mulliken and NPA atomic charges revealed the presence of digital electron-poor and electron-rich atoms (Figure 7b). In general, the NPA atomic charges were higher than the Mulliken atomic charges, while the Mulliken and NPA values confirmed that all the oxygen atoms, as well as the C4, C8, C12, C15, C17, and C18 carbon atoms, acted as nucleophilic sites (Table 3). Accordingly, the electrophilic sites corresponded to the C13, C14, C20, and C21 carbon atoms. Moreover, all hydrogen atoms showed positive charge values, with H5, H10, and H19 being the most electrophilic (Table 3). A high correlation between Mulliken and NPA charge with R = 0.9614 was also observed based on the plot of Figure 7c. It is worth noting that the Mulliken and NPA data were in good agreement with the MEP, XRD packing, and HSA results.

The shape and energy diagram of HOMO and LUMO indicated that the electron donation capacity in the UV region was Δ*E*_HOMO/LUMO_ = 4.266 eV (Figure 8a). Moreover, by measuring the B3LYP/6-311G(d,p) electron transfer in the gaseous state and in MeOH and DMSO (Figure 8b), two broad maxima bands at λ_max_ = 245 and 355 nm were observed, corresponding mainly to HOMO-2-to-LUMO (76%) and HOMO-to-LUMO (96%) transitions, respectively. Similar electron transition results were obtained by experimental UV spectra, as shown in Figure 8c. Specifically, two peaks with λ_max_ at 250 and 350 nm were detected in MeOH and DMSO, which were assigned to π→π* and n→π* electron transitions, respectively. Moreover, no solvatochromism effect was observed by changing the solvents in both the experimental and theoretical studies. The small Δλ shift (~5 nm) between the experimental and DFT data could be attributed to the solute–solvent interaction [32]. In addition, the experimental optical energy band gap (*E*_g_) in MeOH and DMSO was determined using the Tauc equation [33]. Based also on Figure 8d, the direct *E*_g_ value was found to be ~4.51 eV in both solvents, which was close to the Δ*E*_HOMO/LUMO_ value (~4.3 eV).

The GRD quantum parameters of the ligand, including softness (σ), hardness (*η*), chemical potential (*μ*), electrophilicity (*ω*), and electronegativity (*χ*), were also calculated by the following equations (Table 4):*I: Ionization potential = −E_HOMO_*(1)
A: *Electron affinity = −E_LUMO_*(2)
Δ*Ε_gap_*: *Energy gap = E_HOMO_ − E_LUMO_*(3)
*χ*: *Absolute electronegativity* = (*I* + A)/2(4)
*η*: *Global hardness* = (*I* − A)/2(5)
*σ*: *Global softness* = l/*η*(6)
*μ*: *Chemical potential* = − *χ*(7)
*ω*: *Electrophilicity* = *μ*^2^/2*η*(8)

### 3.6. FTIR Investigations 

The experimental FTIR spectrum of 2,5-dimethoxybenzaldehyde in a solid state indicated the presence of several functional groups, which were consistent with its chemical formula. In particular, the peaks at ~3050, 2950-2840, and 1620 cm^−1^ were attributed to the C–H_ph_, C–H_CH3_, and C=O stretching vibrations, respectively (Figure 9a). Moreover, it is clear from Figure 9b that the experimental and DFT-calculated spectra were very similar, while their high compatibility was further confirmed by their excellent correlation with R^2^ = 0.998 (Figure 9c).

### 3.7. Computed and Experimental ^1^H NMR

The experimental ^1^H-NMR spectrum of 2,5-dimethoxybenzaldehyde was recorded in CDCl_3_ (Figure 10a). In the aliphatic region, two broad peaks were detected at 3.65 and 3.76 ppm corresponding to OCH_3_, while the peaks at 6.91 (*d*), 7.11 (*d*), and 7.31 (*s*) ppm were assigned to the three aromatic protons. The aldehyde proton was detected at 10.45 ppm as a singlet. The theoretical NMR-DB [16] (Figure 10b) and GIAO-NMR (Figure 10c) in CDCl_3_ were similar to the experimental spectrum, while the calculated and experimental proton chemical shifts showed a very good correlation, with R^2^ values of 0.9975 and 0.9929, respectively.

### 3.8. Molecular Docking

Both *exo*- and *endo*-isomers of 2,5-dimethoxybenzaldehyde were docked to DNA (PDB ID: 1BNA) under the same level of theory based on the existing data [25]. Interestingly, both isomers showed good and similar docking behaviors and were cross-linked to one DNA helix via two hydrogen bonds to form a (DNA:isomer) complex. No π–π stacking interactions were observed (Figure 11), while the polar 3-OCH_3_ functional group of both isomers did not develop hydrogen-bonding interactions with DNA, as it did not interfere with the DNA helix in the crystal lattice, which supported the minor groove DNA intercalation.

In addition, the binding affinity of the *exo*-isomer indicated its close contact with the DNA surface through a minor groove intercalation mode (Figure 11a) and two short hydrogen bonds to the adenosines of the one DNA helix (Figure 11b). The hydrogen-bonding interactions were assigned to DNA DA17: H∙∙∙OMe (ligand) with 1.984 Å and DNA DA18: H∙∙∙O=C (ligand) with 1.665 Å (Figure 11c). The docking results were consistent with the hydrogen bonds detected in the crystal lattice of the solved structure. In general, the docking effect can be considered a good result when the root mean square deviation (RMSD) value is below 2 Å [32]. The theoretical binding constant (K_b_) and free energy change for the *exo*-isomer were found to be 1.63 × 10^4^ and −5.72 kcal/mol, respectively.

Similar to the *exo*-isomer, the *endo*-isomer was also in contact with DNA via a minor groove intercalation mode (Figure 12a), and one helix binding was observed (Figure 12b). However, this isomer developed two longer hydrogen-bonding interactions with the DNA adenosines, i.e., DA17:H∙∙∙OMe (ligand) with 2.481 Å and DNA DA18:H∙∙∙O=C (ligand) with 1.927 Å (Figure 12c). According to the RMSD, one bond was considered to be a good interaction. The theoretical K_b_ and free energy change were determined at 1.10 × 10^4^ and −5.49 kcal/mol, respectively.

The study showed significant convergence in the docking behavior of both isomers. However, the *exo*-isomer seemed to be slightly more active than the *endo*-isomer, since its hydrogen bonds were stronger, and its binding energy and K_b_ values were higher. These results could be expected, as the *exo*-isomer is more stable, and its structural shape selectivity is more suitable for structure-based drug discovery [16].

### 3.9. Thermal Analysis

The thermal properties of 2,5-dimethoxybenzaldehyde were also evaluated by thermogravimetric–derivative thermogravimetric (TG/DTG) analysis. The TG/DTG curves were obtained at a temperature range of 0–300 °C at a heating rate of 5 °C/min in an open atmosphere (Figure 13). The ligand exhibited acceptable stability up to = 95 °C, while, at temperatures above 95 °C, the ligand was gradually decomposed, and its weight decreased from 100 wt% to 0 wt% via a single broad-step reaction mechanism with T_off_ ≈ 200 °C and complete thermal decomposition.

## 4. Conclusions

In this study, we explored the *exo*⇔*endo* isomerization of 2,5-dimethoxybenzaldehyde based on DFT studies, and the results were compared and confirmed by experimental studies. The formation of the *exo*-isomer was confirmed by the XRD crystallographic analysis structure, while the DFT/XRD structure parameters reflected semi-unity graphical correlations. The hydrogen bonds computed by HSA and MEP analyses were in excellent agreement with the experimental XRD packing results, while the Mulliken and NPA population charge analyses indicated the presence of both nucleophilic and electrophilic sites on the ligand surface. Moreover, the DFT/B3LYP/6-311G(d,p) computational study of the *exo*⇔*endo* isomerization process allowed the identification of the QST2 TS. Furthermore, the TD-SCF/DFT, B3LYP-IR, NMR-DB, and GIAO-NMR calculations were similar to the experimental UV–Vis, direct *E*_g_, FTIR, and ^1^H NMR spectra, respectively. The calculated Mulliken and NPA population charges, along with the HOMO/LUMO and GRD quantum parameters, further supported the *exo*-isomer formation. The ligand also exhibited good thermal stability, with a one-step decomposition mechanism in the range of 100–200 °C. In addition, both isomers showed a very good DNA docking effect, where one helix minor grove with two hydrogen bonds was observed for both isomers. Such compounds can be used in future works as DNA-binding promising drugs.

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
