# Peer review of "Exo⇔Endo Isomerism, MEP/DFT, XRD/HSA-Interactions of 2,5-Dimethoxybenzaldehyde: Thermal, 1BNA-Docking, Optical, and TD-DFT Studies"

_molecules, 2020, doi:10.3390/molecules25245970_

Round 1
Reviewer 1 Report
In this study, the authors explored the exo ⇔ endo isomerization of 2,5-dimethoxybenzaldehyde based on the DFT study, and confirmed the results through experimental studies.
XRD analysis shows that the exo-isomer is considered to be a kinetic preferred isomer. The several hydrogen bond interactions detected in the crystal lattice by XRD are consistent with Hirshfeld surface analysis.
TD-SCF/DFT, B3LYP-IR, NMR-DB and GIAO-NMR calculations are similar to experimental UV-visible, optical energy gap, Fourier transform infrared spectrum and 1H nuclear magnetic resonance spectrum, respectively.
The calculated charges of Mulliken and NPA and the quantum parameters of HOMO / LUMO and GRD further support the formation of exo-isomers.
Both isomers are docked by 1BNA DNA to clarify their binding ability, and the structural parameters calculated by DFT are matched with X-ray diffraction (XRD) crystallographic parameters.
This is a high-quality theoretical and experimental research, so I recommend that it be accepted for publication in the journal Molecules.
Author Response
XRD analysis shows that the exo-isomer is considered to be a kinetic preferred isomer. The several hydrogen bond interactions detected in the crystal lattice by XRD are consistent with Hirshfeld surface analysis.
Thanks Sir
TD-SCF/DFT, B3LYP-IR, NMR-DB and GIAO-NMR calculations are similar to experimental UV-visible, optical energy gap, Fourier transform infrared spectrum and 1H nuclear magnetic resonance spectrum, respectively.
Thank you, this is true by experiment.
The calculated charges of Mulliken and NPA and the quantum parameters of HOMO / LUMO and GRD further support the formation of exo-isomers.
Well Done
Both isomers are docked by 1BNA DNA to clarify their binding ability, and the structural parameters calculated by DFT are matched with X-ray diffraction (XRD) crystallographic parameters.
Thanks Sir
This is a high-quality theoretical and experimental research, so I recommend that it be accepted for publication in the journal Molecules.
Wonderful
Reviewer 2 Report
The manuscript contain a set of some selected spectral data as well as parameters obtained via DFT calculations regarding to rotameric forms of 2,5-dimethoxybenzaldehyde: This is generally a constatation of some facts and data, without scientific discussion. The main goal of this work is not clear for me. I do not see any practical application of the results obtained. I feel, that these-type "database" will be not interesting for potential readers. So, i not recommend this paper for further consideration in 'Molecules'.
Other minor remarks:
- Methodology of localisation and verificetion of transition states should be precisely described.
- The procedure of the optimisation of stationary structures should be described.
- The applications of the level of the theory should be justified.
- Figure 7 and 10. R parameters with three decimal places should be presented instead of R2.
Author Response
The manuscript contain a set of some selected spectral data as well as parameters obtained via DFT calculations regarding to rotameric forms of 2,5-dimethoxybenzaldehyde: This is generally a constatation of some facts and data, without scientific discussion. The main goal of this work is not clear for me. I do not see any practical application of the results obtained. I feel, that these-type "database" will be not interesting for potential readers. So, i not recommend this paper for further consideration in 'Molecules'.
Sorry for that incorrect conclusion.
Methodology of localisation and verificetion of transition states should be precisely described.
2.1. Computational methodology
In order to determine the optimization, Mulliken, NPA, HOMO/LUMO, GRD, TD-SCF/DFT, B3LYP-IR, NMR-DB, and GIAO-NMR quantum-chemical parameters calculations of the desired molecule in gaseous phase have been performed using Becke’s three parameter exchange function (B3) with Lee-Yang-Parr correlation function (LYP) with basis-sets 6-311G(d,p) [19-21]. The DFT/B3LYP/6-311G(d,p) level of theory is found to be very suitable for pure organic compounds like the desired molecule in this study [21]. Moreover, the QTS2 computation method was applied to detect the transition state (TS) of the exo⬄endo isomerization reaction [21].
The procedure of the optimisation of stationary structures should be described.
The DFT/B3LYP/6-311G(d,p) level of theory has been inserted to the text.
The applications of the level of the theory should be justified.
This level of theory is known to be suitable for organic compound less than 22 atomic number.
Figure 7 and 10. R parameters with three decimal places should be presented instead of R2.
Corrected accordingly
Reviewer 3 Report
In this work exo-endo isomerism, MEP/DFT, XRD/HSA-interactions of 2,5-dimethoxybenzaldehyde isomers are described. Both isomers were docked against 1BNA DNA to elucidate their binding ability and the DFT-computed structural parameters were results were matched with the X-ray diffraction crystallographic parameters. XRD analysis showed that the exo-isomer was structurally favored and was also considered as the kinetically preferred isomer. Furthermore, the computed electronic calculations were in good agreement to the experimental UV–Vis, direct Eg, FT–IR, and 1H NMR spectra. I think that the article looks like a short communication and may be published after minor revision.
Notes:
- P. 1, Abstract. The sentence “Both isomers were docked against 1BNA DNA to elucidate their binding ability and the DFT-computed structural parameters were results were matched with the X-ray diffraction (XRD) crystallographic parameters” should be changed by “Both isomers were docked against 1BNA DNA to elucidate their binding ability and the DFT-computed structural parameters results were matched with the X-ray diffraction (XRD) crystallographic parameters”.
- In the Abstract “ultraviolet–visible”, “Fourier transform infrared”, and “1H nuclear magnetic resonance” should be written as “UV-Vis”, “FT-IR” and “1H NMR”, respectively. Generally these abbreviations do not need to be specified in the article (Introduction). These are common abbreviations.
- In the Scheme 1 the dihedral angle of endo-isomer (B) is not correspond to dihedral angle mentioned in the text. It should be checked and corrected.
- Which compound do the UV spectra refer to in the Figure 8c? What concentration the compound has? It should be mentioned in the description to Figure 8c. Also Figures 8a and 8b to which compound are they? It should be mentioned in the descriptions.
- P. 11, line 214. “FT-IR spectra” should be changed by “FT-IR spectrum”.
- Promising application of new obtained results should be added in Conclusions.
- References 4, 7 should be presented in accordance with other references.
Author Response
P. 1, Abstract. The sentence “Both isomers were docked against 1BNA DNA to elucidate their binding ability and the DFT-computed structural parameters were results were matched with the X-ray diffraction (XRD) crystallographic parameters” should be changed by “Both isomers were docked against 1BNA DNA to elucidate their binding ability and the DFT-computed structural parameters results were matched with the X-ray diffraction (XRD) crystallographic parameters”
Corrected accordingly
In the Abstract “ultraviolet–visible”, “Fourier transform infrared”, and “1H nuclear magnetic resonance” should be written as “UV-Vis”, “FT-IR” and “1H NMR”, respectively. Generally these abbreviations do not need to be specified in the article (Introduction). These are common abbreviations.
Corrected
In the Scheme 1 the dihedral angle of endo-isomer (B) is not correspond to dihedral angle mentioned in the text. It should be checked and corrected.
Thank you for this comment the endo-isomer (B) dihedral angle was corrected to 0.
τO1–C2–C3–C4 = 0°
Which compound do the UV spectra refer to in the Figure 8c? What concentration the compound has? It should be mentioned in the description to Figure 8c. Also Figures 8a and 8b to which compound are they? It should be mentioned in the descriptions.
Figure 8. (a) HOMO/LUMO shapes and energy diagram. (b) TD-DFT. (c) Experimental UV spectra, and (d) Optical energy band gap (Eg) of the 2.2x10-6 M of 2,5-dimethoxybenzaldehyde in MeOH and DMSO.
P. 11, line 214. “FT-IR spectra” should be changed by “FT-IR spectrum”.
Corrected
Promising application of new obtained results should be added in Conclusions.
Such compounds can be used in the future work as DNA-binder promising drugs.
References 4, 7 should be presented in accordance with other references.
Corrected
Round 2
Reviewer 2 Report
Basically the structure of the work remained unchanged. I still don't feel that this is an appropriate manuscript since publication in Molecules. At the same time, I can see that two other reviews are rather positive. In the conclusion, in my opinion the manuscript should be evaluated by additional independent reviewer. At this stage, I do not want to prejudge the future of this work.
This manuscript is a resubmission of an earlier submission. The following is a list of the peer review reports and author responses from that submission.